# The Machine-Learning-Mediated Interface of Microbiome and Genetic Risk Stratification in Neuroblastoma Reveals Molecular Pathways Related to Patient Survival

**DOI:** 10.3390/cancers14122874

**Published:** 2022-06-10

**Authors:** Xin Li, Xiaoqi Wang, Ruihao Huang, Andres Stucky, Xuelian Chen, Lan Sun, Qin Wen, Yunjing Zeng, Hansel Fletcher, Charles Wang, Yi Xu, Huynh Cao, Fengzhu Sun, Shengwen Calvin Li, Xi Zhang, Jiang F. Zhong

**Affiliations:** 1Department of Basic Science, School of Medicine, Loma Linda University, Loma Linda, CA 92350, USA; xin.li@usc.edu (X.L.); andres.stucky@usc.edu (A.S.); xuelian.chen@usc.edu (X.C.); hfletcher@llu.edu (H.F.); chwang@llu.edu (C.W.); 2Medical Center of Hematology, Xinqiao Hospital, State Key Laboratory of Trauma, Burn and Combined Injury, Army Medical University, Chongqing 400037, China; xiaoqi.wang@usc.edu (X.W.); ruihao.huang@usc.edu (R.H.); qin.wen@usc.edu (Q.W.); yunjing.zeng@usc.edu (Y.Z.); 3Department of Oncology, Bishan Hospital of Chongqing Medical University, the People’s Hospital of Bishan District, Chongqing 400037, China; lan.sun@usc.edu; 4Divisions of Hematology and Oncology and Regenerative Medicine, Department of Medicine, Loma Linda University, Loma Linda, CA 92350, USA; dyxu@llu.edu (Y.X.); hcao@llu.edu (H.C.); 5Cancer Center of Loma Linda University, Loma Linda, CA 92350, USA; 6Quantitative and Computational Biology Department, University of Southern California, Los Angeles, CA 90089, USA; fengzhu.sun@usc.edu; 7CHOC Children’s Research Institute, Children’s Hospital of Orange County (CHOC), 1201 La Veta Ave., Orange, CA 92868-3874, USA; 8Department of Neurology, University of California—Irvine School of Medicine, 200 S. Manchester Ave. Ste. 206, Orange, CA 92868, USA

**Keywords:** neuroblastoma, genetic risk stratification, machine learning, microbial-based cancer prognosis, RNA-seq, microbial signature

## Abstract

**Simple Summary:**

Neuroblastoma is a highly heterogeneous malignancy with a wide range of outcomes from spontaneous regression to fatal chemoresistant disease, as currently treated according to the risk stratification of the Children’s Oncology Group (COG), resulting in some high COG risk patients receiving excessive treatment, due to lacking predictors for treatment response. Here, we sought to complement COG risk classification by using the tumor intracellular microbiome, which is part of the tumor’s molecular signature. We determine that an intra-tumor microbial gene abundance score, namely M-score, separates the high COG-risk patients into two subpopulations (M_high_ and M_low_) with higher accuracy in risk stratification than the current COG risk assessment, thus sparing a subset of high COG-risk patients from being subjected to traditional high-risk therapies.

**Abstract:**

Currently, most neuroblastoma patients are treated according to the Children’s Oncology Group (COG) risk group assignment; however, neuroblastoma’s heterogeneity renders only a few predictors for treatment response, resulting in excessive treatment. Here, we sought to couple COG risk classification with tumor intracellular microbiome, which is part of the molecular signature of a tumor. We determine that an intra-tumor microbial gene abundance score, namely M-score, separates the high COG-risk patients into two subpopulations (M_high_ and M_low_) with higher accuracy in risk stratification than the current COG risk assessment, thus sparing a subset of high COG-risk patients from being subjected to traditional high-risk therapies. Mechanistically, the classification power of M-scores implies the effect of CREB over-activation, which may influence the critical genes involved in cellular proliferation, anti-apoptosis, and angiogenesis, affecting tumor cell proliferation survival and metastasis. Thus, intracellular microbiota abundance in neuroblastoma regulates intracellular signals to affect patients’ survival.

## 1. Introduction

Cancers emerge from dormant sub-clonal selection switched to mutate, adapt, and grow for spatiotemporally activated dominating sub-clonal expansion [1]. Next-generation sequencing has provided technical tools to explore the genetic landscape of illustrating the intra-tumor genetic heterogeneity of cancer evolutionary patterns [2]. Neuroblastoma is the most common extra-cranial solid tumor occurring in childhood and has diverse clinical presentations, ranging from rapid progression associated with metastatic spread and recurrence to self-limited occupation. About 800 children aged 0 to 14 are diagnosed with neuroblastoma in the United States yearly, and it accounts for 6% of all childhood cancers in the United States [3]. Almost 90% of neuroblastoma patients are children younger than five years old. For pediatric patients with low-risk neuroblastoma, the 5-year overall survival (OS) rate is almost 100%; on the other hand, the 5-year OS is only 50% for those children with high risk. Due to the heterogeneity of tumor biology and invasiveness, the stratification of neuroblastoma risk is vital for medical decision making [4]. Currently, the evaluation criteria before biopsy are imaging and age. A biopsy provides the most accurate and specific evaluation. In the method developed by the Children’s Oncology Group (COG), age at diagnosis, disease stage, tumor histology by the International Neuroblastoma Pathology Classification (INPC) criteria, MYCN status, and DNA ploidy are employed to stratify risk groups [5]. Children in the low-risk group can often be cured with limited treatment, such as surgery alone, whereas those in a high-risk group often need intensive treatments to achieve a cure. The low-risk group has better outcomes, whereas a high COG-risk suggests a poor outcome and will be treated with intensive treatments [6,7,8,9]. There is a need for a more accurate model to help guide medical decisions to avoid excessive treatment. A complementary risk stratification method coupled with genetic landscapes must be established to improve clinical outcomes of neuroblastoma.

Several studies have shown that tissue microbiome was associated with oncogenesis [10]. Traditional microbiome studies focus on extra-cellular microbiome DNA sequences. However, the extra-cellular microbiome (e.g., bacterial and virus) vary with tumor environmental factors such as tissue locations (e.g., oral cavity or gut) and individual difference (e.g., race, sex, age, and nutrients). In contrast, the intracellular microbiome is part of the cancer cell molecular signature [11,12,13], allowing the molecular crosstalk to influence tumor progression. We hypothesize that treatment outcomes may be associated with microbiome alterations, but no studies have linked these changes to treatment. To explore this possibility, we investigate the association of intra-tumor microbial RNA sequences and the survival time of neuroblastoma patients.

Most DNAs in a cell have never been transcribed into RNAs or proteins to alter cellular functions. On the other hand, RNAs (coding or non-coding RNAs) are transcribed in a needed base from human or microbial DNA to participate and alter cellular functions. Consequently, RNAs are a natural selector of functioning human/microbial DNA sequences needed in a cancer cell. Because multiple copies of RNAs, especially those required to alter cellular functions, are transcribed from a single copy of a DNA sequence, RNA-seq can also better detect functional sequences than DNA-seq from the microbiome. Specifically, we demonstrated and optimized an intracellular microbiome prediction score, namely M-score, for high accuracy in risk evaluation. The study design is summarized in Figure 1. Based on the M-score, the high and low M-score subpopulations (M_high_ and M_low_) can be used as neuroblastoma risk stratification to improve neuroblastoma’s clinical outcomes by coupling with the current COG stratification. We further analyzed the related molecular characteristics of M_high_ and M_low_ subpopulations. The *CREB* expression was more activated in the M_high_ subpopulation, directly affecting the expression of *BCL-2*, *VEGF*, *NGF*, and *IGF2* in regulating tumor cell proliferation and survival or metastasis.

## 2. Methods

### 2.1. Datasets and Annotations

The raw RNA-seq data were downloaded from NCI Genomic Data Commons (GDC) Data Portal (https://docs.gdc.cancer.gov, accessed on 1 May 2021) using the GDC data transfer tool available from the National Cancer Institute data portal (https://gdc.cancer.gov/access-data/gdc-data-transfer-tool, accessed on 1 May 2021). A total of 120 neuroblastoma patients with their complete RNA-seq data and clinical information were used in the final analysis. Previous work identified just six contaminants in TCGA (Staphylococcus epidermidis, Propionibacterium acnes, Ralstonia spp., Mycobacterium, Pseudomonas, and Acinetobacter) based on expected low-read abundances across types of cancer [14]. We used *in silico* decontamination methods to remove contaminants, as described in a recent study with the TCGA dataset [13], and eliminated these microbial contaminants from subsequent analyses.

### 2.2. K-Mer Dissimilarity

Alignment-free approaches based on k-mer frequencies have frequently compared metagenomic samples [15]. Compared to alignment-based methods that can only map about 50% of the reads to reference genomes in specific databases [16], alignment-free metagenome comparison methods consider all reads. Thus, we used Skmer, an assembly-free and alignment-free tool, to calculate dissimilarity among different samples using long k-mers (k up to 32) [17]. Since we focused on microbial contributions to neuroblastoma, we first removed reads mapped to all human genes from the RNA-seq data. Therefore, Skmer took the remaining reads after removing human reads from the 120 patients’ RNA-seq as input. The output was a dissimilarity matrix in which each cell represents a pairwise dissimilarity of any pair of the 120 samples. Furthermore, principal coordinates analysis (PCoA) plot was drawn based on the dissimilarity matrix computed by Skmer.

### 2.3. Predicting Survival with Microbial Gene Abundance Using Random Forest Survival Analysis

HUMAnN2 [18] is a tiered fast search method with three search phases. The first phase of screening metagenomic and metatranscriptomic samples to search for known species and then concatenate annotated pangenomes of detected species into a custom sequence dataset. Second, all reads were aligned with the custom dataset built in the first phase using Bowtie 2 [19]. Finally, through a translated search, HUMAnN2 used DIAMOND to align the reads against a protein database, UniRef90 [20]. Based on these three phases, HUMAnN2 can generate the weighted normalized counts of microbial gene abundance in reads per kilobase for each of the 120 samples.

Next, we used the random forest survival analysis in the R package “randomForestSRC” [21] to predict patients’ survival based on the gene abundance profiles with a microbiome prediction score (M-score) [22,23]. The input of the random ForestSRC algorithm was the microbial gene abundance profiles, and the output was the survival time. First, we divided the 120 samples into two datasets, training (70%) and testing (30%). Next, we used all of the microbial gene relative abundance levels computed by HUMAnN2 as features to predict individuals’ survival duration 1,000 times. Finally, Harrell’s C-index (concordance C) measured the average prediction performance, employing the cumulative hazard estimate to the values for comparison. Then, prediction accuracy was always between 0 and 1, which assessed how well the selected model correctly ranked two observations in their observed survival times.

Meanwhile, the predicted values of our developed model were used to define an M-score consisting of 9063 microbial genes of interest. Higher values (M_high_) indicated a higher risk of dying from neuroblastoma and a shorter survival time. We stratified high-risk COG patients into high- and low-risk groups based on the M-score.

### 2.4. Signaling Analysis in the Two Microbiome K-Mers Profile (MKP) Clusters

To understand the underlying mechanisms involved in the superior classification performance of the M-score, we analyzed the related human gene pathways that are differentially expressed between the M_high_ and M_low_ groups. Read counts per gene of human genes for all samples were normalized using Counts per Million (CPM). The average gene expression values of human genes from the two different M risk clusters were used for pathway analysis. We calculated the gene expression ratio of M_high_/M_low_ to assess possible mechanisms. Partek’s Gene Specific Analysis method (Partek^®^ Genomics Suite^®^ software, version 7.0 Copyright ©; 2020 Partek Inc., St. Louis, MO, USA) was used to generate a list of significantly differentially expressed genes between MKP clusters (genes with < 10 reads in any sample were excluded). Significance was determined using a false discovery rate (FDR) adjusted *p*-Value (q-value < 0.05). Ingenuity Pathway Analysis (IPA) software (Qiagen Bioinformatics, Redwood City, CA, USA) was used to analyze gene-specific pathways, and genes enriched in the most significant pathway (lowest *p*-Values) were selected.

## 3. Results

### 3.1. The High-Risk Group of Patients Defined by the COG Criterion

Recent studies [11] showed that the intra-cellular microbiome inside tumor cells impacts clinical outcomes. To investigate the microbial features of neuroblastoma, we analyzed RNA sequencing data, including both human and microbial sequences for 120 neuroblastoma patients recorded in the National Cancer Institute (NCI) Office of Cancer Genomics Therapeutically Applicable Research To Generate Effective Treatments (TARGET) neuroblastoma project (https://ocg.cancer.gov/programs/target, accessed on 1 May 2021). The demographical and clinical characteristics of the 120 patients are summarized in Table 1. In this patient cohort, the mean age at the time of diagnosis is 4.3 years old, and the average survival time of patients who died from neuroblastoma is 1009 days since diagnosis. The majority (80.8%) of patients in the cohort was male, white, without MYCN gene amplification, and classified into a high-risk group according to the COG criterion. The most common primary location of specimens is the abdomen.

### 3.2. Distinct Microbiota Was Found Among Neuroblastoma Patients

After removing reads that could be mapped to the human genome, we performed pattern analysis on the non-human reads, mainly from the microbiome. Skmer [17] extracted k-mer (k = 32) patterns from the non-human sequence for each sample. We named those microbiome patterns Microbiome K-mers Profile (MKP). PCoA [18] took the dissimilarity matrix of MKP computed by Skmer as input to cluster patients by MKP similarity. It showed that the 120 patients were grouped into two MKP clusters, MKP1 and MKP2 (Figure 2). Cox proportional hazards regression analysis (Table 2) showed that the survival times of the patients in these two clusters are significantly different (*p* = 9.505 × 10^−8^). Additionally, it suggested that MKI and COG risk are particularly associated with the patient’s survival time (*p* = 0.05555, *p* = 2.659 × 10^−5^, respectively), while other factors are not statistically significantly associated with the two clusters in our study (Table 2). Notably, MKP is associated with patient survival time with higher significance than all other factors.

It is known that distinct microbial organisms are present in different human body locations among individuals. To investigate whether the two MKP clusters formed due to specimen sources or other clinical factors, we analyzed the association of additional clinical features with the two MKP clusters. The *p*-Value of Pearson’s Chi-squared test indicated that only the COG risk between the two MKP groups was statistically different. However, other factors, including gender, ethnicity, MKI, MYCN status, and location, were not associated with the MKP clusters (Table 3), suggesting that MKP clusters were associated with COG risk but not related to specimen collections. The MKP clusters were not associated with other factors such as gender, ethnicity, tumor location, MKI, and MYCN status either.

We further explored the distributions of patients’ COG risks in the two MKP clusters. Based on the COG risk stratification, children were classified into three different risk groups: low, intermediate, and high. Risk groups were used to help predict the likelihood a child could be cured. However, COG risk stratification is inaccurate, e.g., low COG risk patients could relapse, and high COG risk patients may receive redundant treatment [24]. In our data set, most cases defined as high COG risk patients were distributed in both MKP clusters, but low and intermediate COG risk patients were only presented in the MKP2 cluster (Figure 3). This result suggested that MKP clusters were associated with COG risk, and high COG risk patients could be further classified with MKP as some high COG risk patients were clustered with low and intermediated COG risk patients together in MKP2. Therefore, we compared the survival time of patients in different MKP clusters within various COG risk levels through Cox regression for survival analysis (Table 4). Remarkably, the survival probabilities of high COG risk patients in MKP1 were statistically significantly lower than those of patients with high COG risk in MKP2 (*p* = 6.422 × 10^−6^, hazard ratio = 3.78, Figure 4, Table 4). Moreover, high COG risk patients in MKP2 had significantly lower survival probabilities than patients with low and intermediate COG risk in MKP2 (*p* = 0.0004, hazard ratio = 5.56). In fact, since the stratification for patients with high-risk neuroblastoma is not perfect, the therapy regimen after surgery mainly depends on the experience of physicians. Thus, according to our results, COG risk stratification could be refined more precisely, which is helpful for more accurately assigning the treatment options to improve prognosis. Microbial gene profiles may be employed to classify high COG risk patients into different risk levels precisely.

### 3.3. The Patient’s Survival Time Is Associated with Microbial Gene Abundance

The analysis of clinical features showed a strong association of microbiota with survival time, indicating that the microbiota of neuroblastoma patients could be better used to predict their survival time than COG risk. This result prompted us to use machine-learning methods with patients’ tumor microbiome characteristics to predict the survival probability of neuroblastoma patients. Firstly, we employed HUMAnN2 to calculate the relative abundance of microbial gene families in each tumor sample. The number of microbial gene families in each sample ranged from 1501 to 53,769. Microbial gene families not expressed in more than 80% of patients (i.e., at least 96 patients did not have those gene families) were excluded for further analysis resulting in 9036 microbial gene families. Then, we conducted a random forest using the abundance of 9036 gene families calculated by HUMAnN2 for survival analysis built-in R package ‘randomForestSRC’ to predict patients’ survival time after prioritizing the microbiome features. A randomly selected 70% of patients were used to train a prediction model, and the other 30% were used for validation. The average C-index for the training data is 0.68 and for the validation data is 0.70. The estimated survival function for each individual in our study was shown in Figure 5A (The thick red line is overall ensemble survival; the thick green line is the Nelson–Aalen estimator).

Out-of-bag (OOB) errors were frequently used to evaluate random forests performance. OOB errors were calculated for the samples that were not used for learning the model for each bagged set of samples. Performance assessment results based on OOB errors are shown in Figure 5. Figure 5B shows that individuals with OOB mortality scores above 60% have much shorter survival times than individuals with low OOB mortality scores of less than 0.60. Similarly, about 44 out of 52 individuals (85%) with a survival time longer than 5 years had an OOB mortality score below 20%. In comparison, only 10 out of 68 individuals (15%) with a survival time of fewer than 5 years had a mortality score below 20%. In the OOB Brier Score plot shown in Figure 5C, the red line presents average discrepancies between the observed survival status and the predicted values of patients’ survival time, demonstrating that the prediction model with microbial gene abundance performed pretty well since overall average scores were all below 0.2 at any time. In detail, the average Brier Score increases through the 3 years and then keeps stable after three years. This indicates that our model performed better within a shorter observed period after patients’ diagnosis.

Furthermore, the OOB Continuous Ranked Probability Score (CRPS) (Figure 5D) also demonstrates the MKP model’s excellent performance; the average continuous ranking probability score was below 0.17 all the time. Additionally, the CRPS of sample sets in each risk level, including 0–25%, 25–50%, 50–75%, and 75–100 quantiles, are all lower than 0.25, which is equivalent to guess. For comparison, we also used a random Forest SRC to predict patient survival using other clinical factors mentioned above. The result demonstrated that microbial gene abundance features had superior prediction accuracy than other traditional indices used to estimate neuroblastoma patients’ survival probability (Table 5).

### 3.4. High COG Risk Patients Were Further Separated into High- and Low-Risk Groups with Differential Survival Rates

As mentioned above, we found that high COG risk patients could be stratified into smaller subsets with different survival probabilities according to their various microbial gene abundances. We defined our microbiome prediction score as M-score, applying our prediction model’s predicted values. Higher M-score, M_high,_ means a higher risk for patients dying from neuroblastoma within a specific period. Additionally, the result showed that the distributions of M-scores of high COG risk patients in the training dataset are clustered into two separated groups. The result from Cox regression for survival analysis demonstrated that patients in M_high_ had a higher clinical risk (shorter survival time) than those in M_low_ (*p*-Value = 0.0016). Thus, we set 50 as a threshold of M-score to further classify high COG risk patients into M_high_ or M_low_ risk levels. Patients with M-scores higher than 50 were classified as M_high,_ and those with M-scores lower than 50 were classified as M_low_. Next, we tested the accuracy of the M-score in our validation dataset (Figure 6). Figure 6 indicates that M-score could stratify patients into two M risk groups, in which the survival times of patients were significantly different (*p*-Value = 0.0018). According to the above analysis, the microbiome prediction score based on the M-score can better predict the survival probability of neuroblastoma patients than the current COG method. Cox regression for survival analysis of the training dataset demonstrated that patients in M_high_ had a higher risk of death than those in M_low_ (*p*-Value = 0.0016).

Similarly, in the validation dataset, patients in M_high_ had a higher risk of death than those in M_low_ (*p*-Value = 0.0018). Cox regression for survival analysis of all high-risk patients demonstrated that patients in the M_high_ group had a higher risk of death than those in M_low_ (*p*-Value = 6.422 × 10^−6^). The M-score can better predict patient survival probability (clinical risk) than the current COG method.

### 3.5. The Molecular Crosstalk between Intracellular Microbiota and Tumor Microenvironment Activates CREB and Improves Survival Probability

By analyzing the molecular characteristics of two separate M-score groups, M_high_ and M_low_, we found the *CREB* expression was significantly activated in the M_high_ group. *CREB* is a critical regulator of cell differentiation, proliferation, and survival in cancer cells. We also analyzed the expression of *CREB* target genes involved in anti-apoptosis and found *BCL-2* to be up-regulated in the M_high_ group. Besides, the expression of *VEGF* (M_high_/M_low_ = 1.48 fold, *p*-Value = 0.021), *NGF* (M_high_/M_low_ = 1.79 fold, *p*-Value = 0.005), and *IGF2* (M_high_/M_low_ = 2.12 fold, *p*-Value = 0.014), which were involved in tumor metastasis were also increased in the M_high_ group. Based on the analysis above, we hypothesized that the lower survival probability for patients in the M_high_ group was potentially due to the over-activation of *CREB*, thus inhibiting tumor cell apoptosis and promoting metastasis. The results indicated that *CREB* might be a potential therapeutic target in high-risk neuroblastoma patients (Figure 7).

## 4. Discussion

Microbiota’s responses to cancer treatment reflect intracellular microbiota tumor integration of tumor survival signals. Recent studies have revealed particular microbiome characteristics in several cancers [25]. In particular, the gut microbiome has been shown to have multiple effects on gastrointestinal cancer biology [26]. Thus, study of the intra-tumor microbiome may be an essential step in unveiling the tumor microbiome contributions to cancer progression and improving prognostic prediction. In the present study, we sought to explore microbiome characteristics for risk stratification of neuroblastoma. We characterized a microbiome dissimilarity matrix for 120 patients as MKP profiles and found two groups of neuroblastoma patients with distinct MKP characteristics (MKP1 and MKP2) and survival probability. Since the PCoA results showed that the 120 patients in our study were separated into two MKP clusters, we used Cox proportional hazards regression model to compare the average survival time in these two clusters. The comparison suggested that the two clusters had statistically significant different survival times. Furthermore, Kaplan–Meier curves showed different survival times among patients with high COG risk in these two clusters. This observation prompted us to develop a new method to stratify neuroblastoma patients’ risk more precisely based on their microbial gene expression. Thus, we investigated the association between microbial gene profiles and neuroblastoma patients’ survival time to improve the accuracy of COG risk prediction.

After identifying an association between MKP and COG risk, we set up a machine learning model with microbial profiles to predict survival probability. Our microbiome prediction score (M-score) indicates survival probability accurately coupled with current COG risk stratification criteria, especially high-risk stratification. To minimize the potential effects of contamination, in preparing data for our machine learning model, we utilized a similar method as reported by Poore et al. to remove microbial contamination [13]. We suspect that as neuroblastoma develop, their disorganized, leaky vasculature may allow bacterial translocation, thus forming the specific type of microbiome groups in the tumor tissue of neuroblastoma. The neuroblastoma tumor burden may directly relate to the disorganization of the tumor tissues and microbiome invasion. Alternatively, the invasion of the microbiome may promote an inflammatory reaction and suppression of immune response that worsens prognosis. Over 9190 intra-tumor bacterial species were reported recently [11]. Whether or not the microbial profile of neuroblastoma plays a causal role in tumorigenesis remain elusive. However, it is of interest to further explore effects of intra-tumor microbiome on the immune system and its interactions with tumor cells.

Microbiome metabolic pathways were recently reported to associate with the response to immunotherapy [11]. To investigate why the M-scores displayed better performance in risk classification than COG risk stratification, we calculated M_high_/M_low_ gene expression ratios. We identified the over-activation of *CREB* and its regulating genes, including *BCL-2* [27], *VEGF*, *NGF*, and *IGF-2* [28], in M_high_ patients. *CREB* plays an essential role in various biological functions, including cellular proliferation, differentiation, and adaptive responses in the neuronal system [29,30]. Recently, accumulating evidence showed that *CREB* participates in the regulation of immortalization and transformation of cancer cells, including prostate cancer, breast cancer, non-small-cell lung cancer, and acute leukemia [27,31,32,33]. Our study also demonstrates that the microbiome could influence the tumor microenvironment. The deeper mechanism underlying this effect needs further exploration. The intra-tumor microbial signatures of neuroblastoma we reported here is an intrinsic part of the neuroblastoma molecular signature. Although the pathology of those microbial signatures remains elusive, such microbial molecular signatures could be used to improve the diagnosis and prognosis of neuroblastoma via refining current risk stratification.

## 5. Conclusions

We applied the machine-learning-mediated interface between the microbiome and genetic risk stratification of neuroblastoma patients defined by the Children’s Oncology Group (COG) to discover molecular pathways related to patient survival.

## Figures and Tables

**Figure 1 cancers-14-02874-f001:**
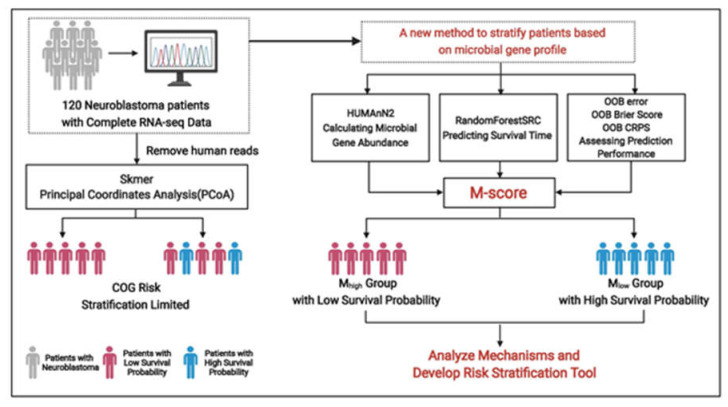
Summary of study design. Machine learning was developed to overcome COG risk stratification’s limitation for identifying patients with high survival probability in the high COG-risk group. The algorithm determines the intratumoral and intracellular microbial gene abundance score, namely M-score, to separate the high COG-risk patients into two subpopulations (M_high_ and M_low_) with higher accuracy in risk stratification and is complementary to the current COG risk assessment, thus sparing a subset of high COG-risk patients from being subjected to traditional high-risk therapies.

**Figure 2 cancers-14-02874-f002:**
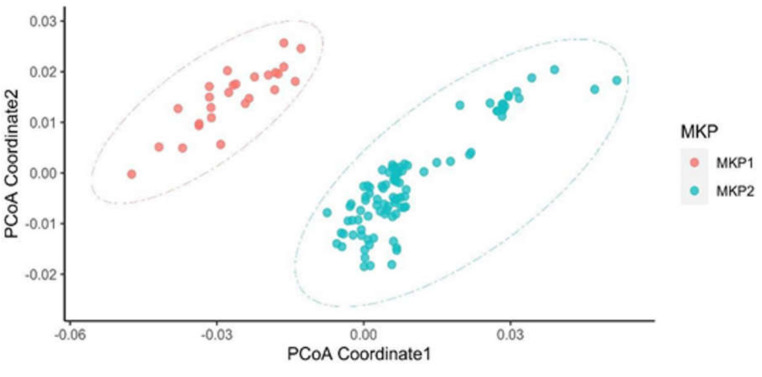
Principal coordinate analysis of the gene dissimilarity matrix computed by Skmer (MKP). Based on microbial sequence similarity, 120 neuroblastoma patients were grouped into two MKP clusters, which were defined as MKP1 and MKP2. Patients in these two groups had significantly different microbial profiles in their tumor tissues. The survival probability of patients in MKP1 was statistically lower than that of patients in MKP2 (*p* = 9.505 × 10^−8^).

**Figure 3 cancers-14-02874-f003:**
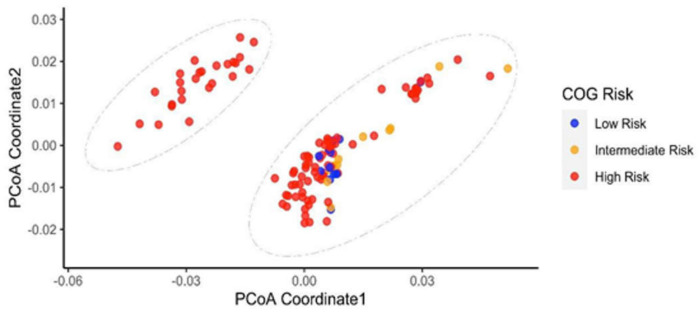
Principal coordinate analysis of the gene dissimilarity matrix computed by Skmer (COG Risk). One of the microbiome clusters, MKP2, contained patients defined by COG criteria as high, medium, and low risk. The COG high-risk patients had distinct microbiome characteristics. Some COG high-risk patients in MKP2 had similar microbiome features to those with COG medium and low risk. Red, orange, and blue points represent patients categorized by COG criteria as high, intermediate, and low risk. Remarkably, all patients clustered in MKP1 were COG high-risk; however, MKP2 contained patients in all three different COG risk levels.

**Figure 4 cancers-14-02874-f004:**
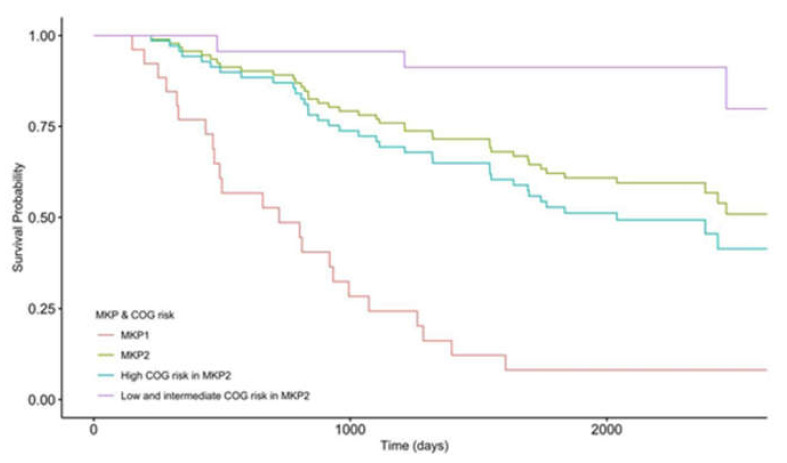
Kaplan-Meier estimator within four different MKP and COG risk groups. The COG high-risk patients in MKP1 and MKP2 had lower survival probabilities than COG low and intermediate-risk patients in MKP2. The hazard ratio (HR) for death among high-risk patients in MKP1 was 17.1 times that of patients with low and intermediate COG risk in MKP2 (*p* = 4.605 × 10^−9^). The HR for death among high-risk patients in MKP2 was 5.56 times that of patients with low and intermediate-risk in MKP2 (*p* = 0.0004). However, the survival probability of high-risk patients in MKP1 was lower than that of patients with high risk in MKP2. The HR for death among high-risk patients in MKP1 was 3.78 times that of those in MKP2 (*p* = 6.422 × 10^−6^). Additionally, the total survival probability for patients in MKP1 was lower than those in MKP2. The HR for death in MKP1 was 5 times that in MKP2 (*p* = 9.505 × 10^−8^).

**Figure 5 cancers-14-02874-f005:**
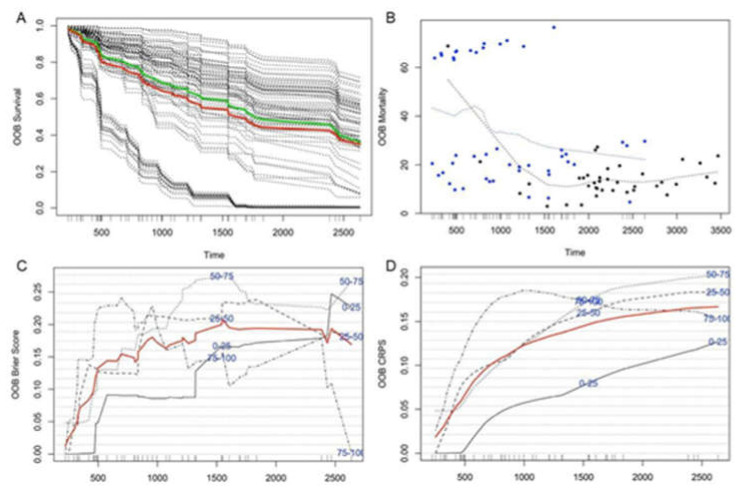
Performance measures for the MKP model. (**A**) The estimated survival function for each individual. The thick red line represents overall ensemble survival, and the thick green line represents the Nelson–Aalen estimator. (**B**) The plot of survival probabilities is estimated for each patient based on our prediction model in the OOB ensemble (points in blue correspond to death events; black points are censored observations). (**C**) OOB time-dependent Brier Score (0 = perfect, 1 = poor, and 0.25 = guessing). The score is shown stratified by ensemble mortality into four groups corresponding to the 0–25, 25–50, 50–75, and 75–100 percentile values of mortality. The red line is the overall (non-stratified) time-dependent Brier score. (**D**) OOB time-dependent CRPS (0 = perfect, 1 = poor, and 0.25 = guessing). The score is shown stratified by ensemble mortality into four groups corresponding to the 0–25, 25–50, 50–75, and 75–100 percentile values of mortality. The red line is the overall (non-stratified) time-dependent CRPS.

**Figure 6 cancers-14-02874-f006:**
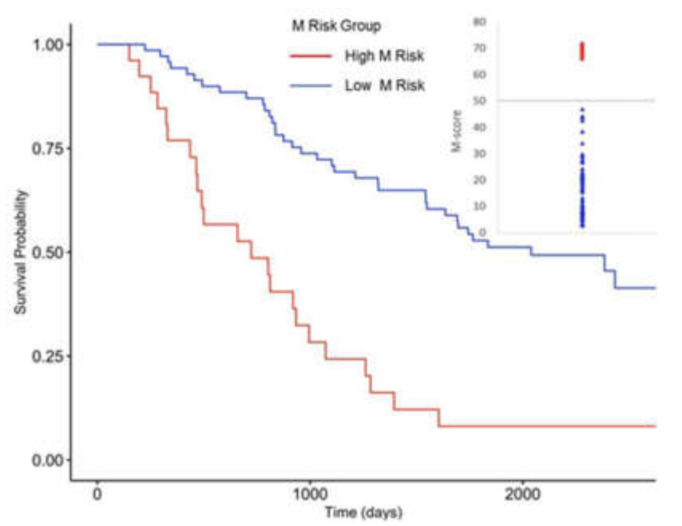
Predicting Patients’ Survival Probability with Microbial Gene Abundance. Distributions of M-scores in COG high-risk patients were shown in the inset (upper right). Patients with M-scores higher than 50 were classified as M_high_, and those with M-scores lower than 50 were classified as M_low_. Survival analysis indicated that M_high_ patients have significantly lower survival probability (*p*-Value = 6.422 × 10^−6^). Line and points in red represent M_high_; lines in blue represent M_low_.

**Figure 7 cancers-14-02874-f007:**
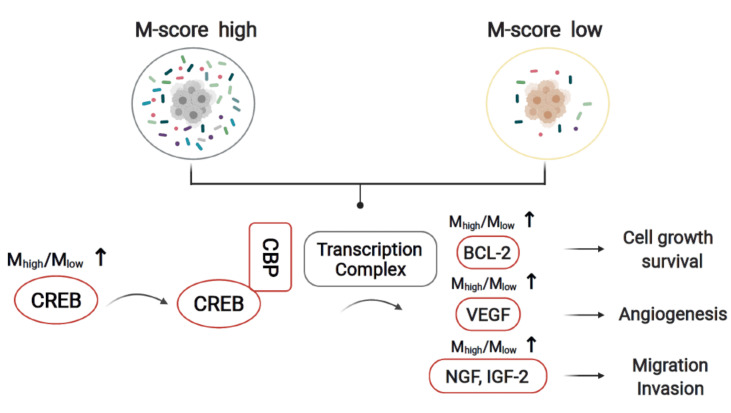
*CREB* activation may be a key genetic event related to M-score that contributes to the tumorigenesis in the M_high_ group with low-survival probability. Among the 120 patients, the *CREB* was over activated in the M_high_ group relative to the M_low_ group. This could be responsible for activating downstream genes related to cell growth, survival, angiogenesis, migration, and invasion, including *BCL-2*, *VEGF*, *NGF*, and *IGF-2*, thus leading to the lower survival probability in the M_high_ group.

**Table 1 cancers-14-02874-t001:** Characteristics of 120 neuroblastoma patients.

Characteristics	N (%)
Gender	
Male	70 (58.3)
Female	50 (41.7)
Ethnicity	
White	85 (70.8)
Others	35 (29.2)
MKI	
Low	35 (29.2)
Intermediate	34 (28.3)
High	26 (21.7)
Unknown	25 (20.8)
MYCN Status	
Amplified	23 (19.2)
Not Amplified	96 (80)
Unknown	1 (0.8)
COG Risk	
Low Risk	12 (10.0)
Intermediate Risk	11 (9.2)
High Risk	97 (80.8)
Location of tumor	
Abdomen	104 (86.7)
Others	16 (13.3)
	**Mean (SD)**
Age(in years)	4.3 (2.5)
Survival Time(in days)	
Event	1009.2 (617.2)
Censored	2204.5 (734.5)

**Table 2 cancers-14-02874-t002:** Cox proportional hazards regression model test result.

Variables	*p*-Value
MKP Clusters	9.505 × 10^−8^
Gender	0.6899
MKI	0.0556
MYCN Status	0.2449
COG Risk	2.659 × 10^−5^
Location	0.9878
Ethnicity	0.5443

**Table 3 cancers-14-02874-t003:** Chi-square test of independence between MKP clusters and other potential factors.

Variables	Chi-Square (df)	*p*-Value
Gender	0.0898(1)	0.7645
Ethnicity	0.1997(1)	0.655
MKI	5.0892(3)	0.1654
MYCN Status	0.6865(1)	0.4074
COG Risk	7.8701(2)	0.0195
Location of tumor	0.0005(1)	0.9827

**Table 4 cancers-14-02874-t004:** *p*-Values and hazard ratios between different risk groups in MKP1 and MKP2.

Variables	*p*-Value	Hazard Ratio
MKP1 vs. MKP2	9.505 × 10^−8^	5
MKP1 vs. COG high risk in MKP2	6.42210^−6^	3.78
MKP1 vs. COG low and intermediate risk in MKP2	4.60510^−9^	17.1
MKP2 vs. COG high risk in MKP2	0.2119	0.75
MKP2 vs. COG low and intermediate risk in MKP2	0.0041	4.07
COG high risk in MKP2 vs. COG low/intermediate risk in MKP2	0.0004	5.56

**Table 5 cancers-14-02874-t005:** Error rate comparison with different features.

Variables	Error Rate (%)
**Microbial Gene Abundance**	29.87
**Gender**	71.67
**MKI**	53.65
**MYCN Status**	75.21
**COG Risk**	68.97
**Location of tumor**	82.39

## Data Availability

The datasets used and analyzed during the current study are available from the corresponding author on reasonable request.

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
