# Peer review of "The Machine-Learning-Mediated Interface of Microbiome and Genetic Risk Stratification in Neuroblastoma Reveals Molecular Pathways Related to Patient Survival"

_cancers, 2022, doi:10.3390/cancers14122874_

Round 1
Reviewer 1 Report
This is a very interesting analysis looking at how the intestinal microbiome impacts risk classification in neuroblastoma. It seems that the algorithm might be skewed by the inclusion of the low and intermediate risk patients as they did not cluster independent of the high-risk patients.
It also seems to be a jump to me to suggest that the microbiome can identify patients who receive too much treatment. Would the high risk patients who are clustered with low and intermediate risk maintain better outcomes if treatment was reduced? Also, are outcomes sufficiently good for those with more favorable outcomes to reduce treatment?
Are there any treatments that can be used to adjust the microbiome to shift to a more favorable profile? Is this a modifiable factor or a static representation of disease? How do you envision this type of information being utilized clinically?
Overall, I think it is an interesting approach to optimization of risk stratification. I would like to see some refining of the discussion and approach to the use of this type of information.
Reviewer 2 Report
The study entitled The machine-learning-mediated interface between the microbiome and genetic risk stratification in Neuroblastoma regulates patients intracellular signals responding to treatment for improved survival by Xin Li and colleagues explored tumor intracellular microbiome using machine learning approaches and coupled it to neuroblastoma risk stratification groups to predict better clinical outcome in neuroblastoma patients. The manuscript is well designed and well written. My comments are below:
Which microbial species are more predominant in the genome of neuroblastoma patients?
If this statement is true in Neuroblastoma then, why microbial DNA integration into the human genome is more in tumors than in normal human tissue.
What is the status of ALK and PHOX2B mutations in M high vs. M low tumors?
Did the authors look at the immune microenvironment components in M high vs. M low tumors?
Reviewer 3 Report
Comments for “The machine-learning-mediated interface between the microbiome and genetic risk stratification in Neuroblastoma regulates patients’ intracellular signals responding to treatment for improved survival”
Major:
- The description of the method part is confusing for readers. a. authors mentioned the containments in the sample, so why the containments happen, how to minimize the effect of containment, and what is the result if the containments information were included in the data. b. an appropriate reference for “HUMAnN2” should be added in the manuscript, the bowtie2 is a commonly used method for sequencing alignment, what is the relationship between bowtie2 and HUMAnN2?
- What is the input data for the Random Forest Survival Analysis?
- If there is more than one study reported the RNA-seq data from neuroblastoma patients, authors should use another dataset to validate their results
- The meaning of out-of-bag should be clarified.
Minor:
- The font should be consensus in the manuscript (such as page 2 row 74)
- The quality of figures should be improved though.
Round 2
Reviewer 3 Report
Authors have addressed my early concerns